# COFS
## CONTROLLABLE FURNITURE LAYOUT SYNTHESIS

## ABSTRACT

Realistic, scalable, and controllable generation of furniture layouts is essential for many applications in virtual reality, augmented reality, game development and synthetic data generation. The most successful current methods tackle this problem as a sequence generation problem which imposes a specific ordering on the elements of the layout, making it hard to exert fine-grained control over the attributes of a generated scene. Existing methods provide control through *object-level conditioning*, or scene completion, where generation can be conditioned on an arbitrary subset of furniture objects. However, *attribute-level conditioning*, where generation can be conditioned on an arbitrary subset of object attributes, is not supported. We propose COFS, a method to generate furniture layouts that enables fine-grained control through attribute-level conditioning. For example, COFS allows specifying only the scale and type of objects that should be placed in the scene and the generator chooses their positions and orientations; or the position that should be occupied by objects can be specified and the generator chooses their type, scale, orientation, etc. Our results show both qualitatively and quantitatively that we significantly outperform existing methods on attribute-level conditioning.

## 1 INTRODUCTION

Automatic generation of realistic assets enables content creation at a scale that is not possible with traditional manual workflows. It is driven by the growing demand for virtual assets in both the creative industries, virtual worlds, and increasingly data-hungry deep model training. In the context of automatic asset generation, 3D scene and layout generation plays a central role as much of the demand is for the types of real-world scenes we see and interact with every day, such as building interiors.

Deep generative models for assets like images, videos, 3D shapes, and 3D scenes have come a long way to meet this demand. In the context of 3D scene and layout modeling, in particular auto-regressive models based on transformers enjoy great success. Inspired by language modeling, these architectures treat layouts as sequences of tokens that are generated one after the other and typically represent attributes of furniture objects, such as the type, position, or scale of an object. These architectures are particularly well suited for modeling spatial relationships between elements of a layout. For example, (Para et al., 2021) generate two-dimensional interior layouts with two transformers, one for furniture objects and one for spatial constraints between these objects, while SceneFormer (Wang et al., 2021) and ATISS (Paschalidou et al., 2021) extend interior layout generation to 3D.

A key limitation of a basic autoregressive approach is that it only provides limited control over the generated scene. It enforces a sequential generation order, where new tokens can only be conditioned on previously generated tokens and in addition it requires a consistent ordering of the token sequence. This precludes both *object-level conditioning*, where generation is conditioned on a partial scene, e.g., an arbitrary subset of furniture objects, and *attribute-level conditioning*, where generation is conditioned on an arbitrary subset of attributes of the furniture objects, e.g., class or position of target objects. Most recently, ATISS (Paschalidou et al., 2021) partially alleviates this problem by randomly permuting furniture objects during training, effectively enabling *object-level conditioning*. However, attribute-level conditioning still remains elusive.

We aim to improve on these results by enabling attribute-level conditioning, in addition to object-level conditioning. For example, a user might be interested to ask for a room with a table and two chairs, without specifying exactly where these objects should be located. Another example is to perform

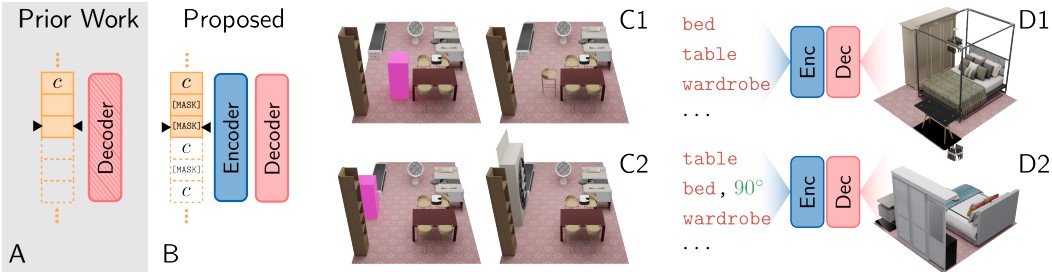

Figure 1: **Motivation.** Current autoregressive layout generators (**A**) provide limited control over the generated result, since any generated value (denoted by black triangles) can only be conditioned on values that occur earlier in the sequence (values that are given as condition are denoted with $c$). Our proposed encoder-decoder architecture (**B**) adds bidirectional attention through an encoder, allowing the model to *look ahead*, so that all values in the sequence can be given as condition. This enables conditioning on an arbitrary subset of objects or object attributes in a layout. In **C1, C2** only the position of an object, shown as pink cuboid, is given as condition and COFS performs context-aware generation of the remaining attributes. In **D1**, only object types are provided as condition, and **D2** adds the bed orientation to the condition. Note how the layout adapts to fit the updated condition.

object queries for given geometry attributes. The user could specify the location of an object and query the most likely class, orientation, and size of an object at the given location. Our model thereby extends the baseline ATISS with new functionality while retaining all its existing properties and performance.

The main technical difficulty in achieving attribute-level conditioning is due to the autoregressive nature of the generative model. Tokens in the sequence that define a scene are generated iteratively, and each step only has information about the previously generated tokens. Thus, the condition can only be given at the start of the sequence, otherwise some generation steps will miss some of the conditioning information. The main idea of our work is to allow for attribute-level conditioning using two mechanisms: (i) Like ATISS, we train our generator to be approximately invariant to object permutations by randomly permuting furniture objects at training time. This enables object-level conditioning since an arbitrary subset of objects can be given as the start of the sequence. To condition on a partial set of object attributes however, the condition is not restricted to the start of the sequence. Attributes that are given as condition follow unconstrained attributes that need to be generated. (ii) To give our autoregressive model knowledge of the entire conditioning information in each step, we additionally use a transformer encoder that provides cross-attention over the complete conditioning information in each step. These two mechanisms allow us to accurately condition on arbitrary subsets of the token sequence, for example, only on tokens corresponding to specific object attributes.

In our experiments, we demonstrate four applications: (i) attribute-level conditioning, (ii) attribute-level outlier detection, (iii) object-level conditioning, and (iv) unconditional generation. We compare to three current state-of-the-art layout generation methods (Ritchie et al., 2019; Wang et al., 2021; Paschalidou et al., 2021) and show performance that is on par or superior on unconditional generation and object-level conditioning, while also enabling attribute-level conditioning, which, to the best of our knowledge, is currently not supported by any existing layout generation method.

## 2 RELATED WORK

We discuss recent work that we draw inspiration from. In particular, we build on previous work in Indoor Scene Synthesis, Masked Language Models, and Set Transformers.

**Indoor Scene Synthesis**: Before the rise of deep-learning methods, indoor scene synthesis methods relied on layout guidelines developed by skilled interior designers, and an optimzation strategy such that the adherence to those guidelines is maximized (Yu et al., 2011; Fisher et al., 2012; Weiss et al., 2019). Such optimization is usually based on sampling methods like simulated annealing, MCMC, or rjMCMC. Deep learning based methods, e.g. (Wang et al., 2019; Ritchie et al., 2019; Wang et al., 2021; Paschalidou et al., 2021) are substantially faster and can better capture the variability of the design space. The state-of-the-art methods among them are autoregressive in nature. All of these operate on a top-down view of a partially generated scene. PlanIT and FastSynth then autoregressively generate the rest of the scene. FastSynth uses separate CNNs+MLPs to create probability distributions over location, size and orientation and categories. PlanIT on the other hand generates graphs where

nodes are objects and edges are constraints on those objects. Then a scene is instantiated by solving a CSP on that graph.

Recent methods, SceneFormer (Wang et al., 2021) and ATISS (Paschalidou et al., 2021) use transformer based architectures to sidestep the problem of rendering a partial scene which makes PlanIT and FastSynth slow. This is because using a transformer allows the model to accumulate information from previously generated objects using the attention mechanism. SceneFormer flattens the scene into a structured sequence of the object attributes, where the objects are ordered lexicographically in terms of their position. It then trains a separate model for each of the attributes. ATISS breaks the requirement of using a specific order by training on all possible permutations of the object order and removing the position encoding. In addition, it uses a single transformer model for all attributes and relies on different decoding heads which makes it substantially faster than other models while also having significantly fewer parameters.

**Masked Language Models**: Masked Language Models (MLMs) like BERT (Devlin et al., 2019), ROBERTa (Liu et al., 2019), and BART (Lewis et al., 2020) have been very successful in pre-training for language models. These models are pretrained on large amounts of unlabeled data in an unsupervised fashion, and are then fine-tuned on a much smaller labeled dataset. These fine-tuned models show impressive performance on their corresponding downstream tasks. However, the generative capability of these models has not been much explored except by Wang et al. in (Wang & Cho, 2019), which uses a Gibbs-sampling approach to sample from a pre-trained BERT model. Follow up work in Mansimov et al. (Mansimov et al., 2020), proposes more general sampling approaches. However, the sample quality is still inferior to autoregressive models like GPT-2 (Radford et al., 2019) and GPT-3 (Brown et al., 2020). More recently, MLMs have received renewed interest especially in the context of image-generation (Issenhuth et al., 2021; Chang et al., 2022). MaskGit (Chang et al., 2022) shows that with a carefully designed masking schedule, high quality image samples can be generated from MLMs with parallel sampling which makes them much faster than autoregressive models. Edi-BERT (Issenhuth et al., 2021) shows that the BERT masking objective can be succesfully used with a VQGAN (Esser et al., 2021) representation of an image to perform high quality image editing. Our model most closely resembles BART when used as a generative model.

**Set Transfomers**: Zaheer et al. (Zaheer et al., 2017) introduced a framework called DeepSets providing a mathematical foundation for networks operating on set-structured data. A key insight is that operations in the network need to be permutation invariant. Methods based on such a formulation were extremely successful, especially in the context of point-could processing (Charles et al., 2017; Ravanbakhsh et al., 2016). Transformer models without any form of positional encoding are permutation invariant by design. Yet, almost all the groundbreaking works in transformers use some from of positional encoding, as in objection detection (Carion et al., 2020), language generation (Radford et al., 2019; Brown et al., 2020), and image-generation (Chang et al., 2022). One of the early attempts to use a truly permutation invariant set transformer was in Set Transformer (Lee et al., 2019), who methodically designed principled operations that are permutation invariant but could only achieve respectable performance in toy-problems. However, recent work based on (Lee et al., 2019) shows impressive performance in 3d-Object Detection (Chenhang He & Zhang, 2022), 3d Pose Estimation (Ugrinovic et al., 2022), and SFM (Moran et al., 2021).

## 3 METHOD

Our goal is to design a generative model of object layouts that allows for both object-level and attribute-level conditioning. Attribute-level conditioning enables more flexible partial layout specification, for example specifying only the number and types of objects in a layout, but not their positions, or exploring suggestions for plausible objects at given positions in the layout. An overview of our architecture is given in Figure 2.

### 3.1 LAYOUT REPRESENTATION

We focus on 3D layouts in our experiments. A 3D layout $\mathcal{L} = (\mathcal{I}, \mathcal{B})$ is composed of two elements - a top-down representation of the layout boundary $\mathcal{I}$, such as the walls of a room, and a set of $k$ three-dimensional oriented bounding-boxes $\mathcal{B} = \{B_i\}_{i=1}^{k}$ of the objects in the layout. The boundary is given as a binary raster image and each bounding box is represented by four attributes: $B_i = (\tau_i, t_i, e_i, r_i)$, representing the object class, center position, size, and orientation, respectively. The orientation is a rotation about the up-axis, giving a total of 8 scalar values per bounding box.

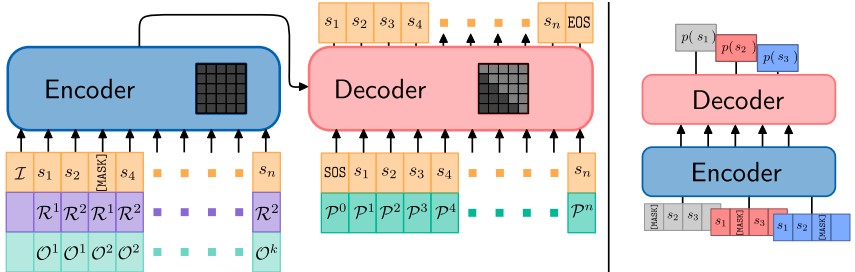

Figure 2: **COFS Overview**. (**Left**): The model is a BART-like encoder-decoder model, with bidirectional attention in the encoder and an autoregressive decoder. The encoder encodes the layout as a set without ordering information and therefore does not receive (*absolute*) position tokens. However, to disambiguate a single object, the encoder receives additional information in the form of *Relative Position Tokens* $\mathcal{R}^i$, and the *Object Index Tokens* $\mathcal{O}^i$. During training, object order is randomly permuted in a layout and a random proportion of tokens is replaced with a [MASK] token. The decoder outputs a sequence representation of the set and is trained with *Absolute Position Tokens* $\mathcal{P}^i$. It performs two tasks - 1. copy-paste: the decoder copies the unmasked attributes to their proper location 2: mask-prediction: the decoder predicts the actual value of the token corresponding to a [MASK] token in the encoder input. (**Right**): During inference, to measure likelihood, we create a copy of the sequence with each token masked out. The decoder outputs a probability distribution over the possible values of the masked tokens.

The layout is arranged into a layout sequence $S$ by concatenating all bounding box parameters. Additionally, special start and stop tokens SOS and EOS are added to mark the start and the end of a sequence: $S = [\text{SOS}; B_1; \ldots; B_k; \text{EOS}]$, where $[;]$ denotes concatenation. The layout boundary $\mathcal{I}$ is not generated by our method, but it is used as condition, Section 3.3 provides details.

## 3.2 GENERATIVE MODEL

We use a transformer-based generative model, as these types of generative models have shown great performance in the current state of the art. Originally proposed as a generative model for language, transformer-based generative models represents layouts as a sequence of tokens $S = (s_1, \ldots, s_n)$ that are generated auto-regressively; one token is generated at a time, based on all previously generated tokens:

$$p(s_i|S_{<i}) = f_\theta(S_{<i}), \tag{1}$$

where $p(s_i|S_{<i})$ is the probability distribution over the value of token $s_i$, computed by the generative model $f_\theta$ given the previously generated tokens $S_{<i} = (s_1, \ldots, s_{i-1})$. We sample from $p(s_i|S_{<i})$ to obtain the token $s_i$. Each token represents one attribute of an object, and groups of adjacent tokens correspond to objects. More details on the layout representation are described in Section 3.1.

**Limitations of traditional conditioning.** To condition a transformer-based generative model on a partial sequence $C = (c_0, \ldots, c_m)$, we can replace tokens of $S$ with the corresponding tokens of $C$, giving us the sequence $S^C$. This is done after each generation step, so that the probability for the token in each step is conditioned on $S^C_{<i}$ instead of $S_{<i}$:

$$p(s_i|S^C_{<i}) = f_\theta(S^C_{<i}). \tag{2}$$

Each generated token $s_i$ in $S^C$ (i.e. tokens that are not replaced by tokens in $C$) needs to have knowledge of the full condition during its generation step, otherwise the generated value may be incompatible with some part of the condition. Therefore, since each generated token $s_i$ only has information about the partial sequence $S^C_{<i}$ of tokens that are closer to the start of the sequence, the condition can only be given as start of the sequence:

$$S^C = \begin{cases} c_i \text{ if } i \leq |C| \\ s_i \text{ otherwise.} \end{cases} \tag{3}$$

Typically both the objects and the attributes of the objects in the sequence are consistently ordered according to some strategy, for example based on a raster order of the object positions (Para et al., 2021), or on the object size (Wang et al., 2019). Therefore, a generative model $f_\theta^{\text{ordered}}$ that is only trained to generate sequences in that order cannot handle different orderings, so that in general:

$$f_\theta^{\text{ordered}}(S^C_{<i}) \neq f_\theta^{\text{ordered}}(\pi_o(S^C_{<i})), \tag{4}$$

where $\pi_o$ is a random permutations of the objects in sequence $S_{<i}$. The consistent ordering improves the performance of the generative model, but also presents a challenge for conditioning: it limits the information that can appear in the condition. In a bedroom layout, for example, if beds are always generated before nightstands in the consistent ordering, the layout can never be conditioned on nightstands only, as this would preclude the following tokens from containing a bed.

**Object-level conditioning.**   Recent work (Paschalidou et al., 2021) tackles this issue by forgoing the consistent object ordering, and instead training the generator to be approximately invariant to permutations $\pi_o$ of objects in the sequence:

$$f_\theta(S_{<i}^C) \approx f_\theta(\pi_o(S_{<i}^C)), \tag{5}$$

This makes generation more difficult, but enables object-level conditioning by allowing conditioning on *arbitrary* subset of objects, as now arbitrary objects can appear at the start of the sequence. However, since only objects are permuted and not their attributes, it does not allow conditioning on subsets of object attributes. Permuting object attributes to appear at arbitrary positions in the sequence is not a good solution to enable attribute-level conditioning, as this would make it very hard for the generator to determine which attribute corresponds to which object.

**Attribute-level conditioning.**   We propose to extend previous work to allow for attribute-level conditioning by using two different conditioning mechanisms, in addition to the approximate object permutation invariance: First, similar to previous work, we provide the condition as partial sequence $C$. However, unlike previous work, some tokens in the condition are unconstrained and will not be used to replace generated tokens. We introduce special mask tokens $\mathcal{M}$ in $C$ to mark these unconstrained tokens. For example, if all tokens corresponding to object positions and orientations in $C$ are mask tokens, the positions and orientations will be generated by our model, only the remaining tokens: object types and sizes will be constrained by the condition. The constrained sequence $S^C$, is then defined as:

$$S^C = \begin{cases} c_i \text{ if } i \leq |C| \text{ and } c_i \neq \mathcal{M} \\ s_i \text{ otherwise.} \end{cases} \tag{6}$$

Second, to provide information about the full condition to each generated token, we modify $f_\theta$ to use a transformer encoder $g_\phi$ that encodes the condition $C$ into a set of feature vectors that each generated token has access to:

$$p(s_i|S_{<i}^C, C) = f_\theta(S_{<i}^C, C^g) \text{ where } C^g = \{g_\phi(c_1, C), \ldots, g_\phi(c_{|C|}, C)\}, \tag{7}$$

where $C^g$ is the output of the encoder, a set of encoded condition tokens. We use a standard transformer encoder-decoder setup (Vaswani et al., 2017) for $f_\theta$ and $g_\phi$, implementation details are provided in Section 3.3, and the complete architecture is described in detail in the appendix.

**Parameter probability distributions.**   The generative model outputs a probability distribution over one scalar component of the bounding box parameters in each step. Probability distributions over the discrete object class $\tau$ are represented as vectors of logits $l_\tau$ over *discrete* choices that can be converted to probabilities with the softmax function. Similar to previous work (Paschalidou et al., 2021; Salimans et al., 2017), we represent probability distributions over *continuous* parameters, like the center position, size, and orientation, as mixture of $T$ logistic distributions.

$$p(b) = \frac{1}{\sum_i \pi_i} \sum_{i=1}^{T} \alpha_i \text{Logistic}(\mu_i, \sigma_i), \qquad p(\tau) = \text{softmax}(l_\tau), \tag{8}$$

where $b$ is a single scalar attribute from $t_i$, $e_i$, or $r_i$. The mixture weight, mean and variance of the logistic distribution components are denoted as $\alpha$, $\mu$, $\sigma$, respectively. Each probability distribution over a continuous scalar component is parameterized by a $3T$-dimensional vector, and probability distributions over the object class are represented as $n_\tau$-dimensional vectors, where $n_\tau$ is the number of object classes.

### 3.3 Implementation

**Condition encoder** $g_\phi$: To encode the condition $C$ into a set of encoded condition tokens $C^g$, we use a Transformer encoder with full bidirectional attention. As positional encoding, we provide two additional sequences: *object index tokens* $\mathcal{O}^i$ provide for each token the object index in the permuted sequence of objects; and *relative position tokens* $\mathcal{R}^i$ provide for each token the element index inside

the attribute tuple of an object. Since the attribute tuples are consistently ordered the index can be used to identify the attribute type of a token. These sequences are used as additional inputs to the encoder. The encoder architecture is based on BART (Lewis et al., 2020), details are provided in the appendix.

**Boundary encoder** $g_\psi^\mathcal{I}$: To allow conditioning on the layout boundary $\mathcal{I}$, we prepend a feature vector encoding $z_\mathcal{I}$ of the boundary to the input of the condition encoder, as shown in Figure 2, so that the encoder receives both $z_\mathcal{I}$ and the condition sequence $C$. Similar to ATISS, we use an untrained ResNet-18 (He et al., 2016) to encode a top-down view of the layout boundary into an embedding vector.

**Generative model** $f_\theta$: The generative model is implemented as a Transformer decoder with a causal attention mask. Each block of the decoder performs cross-attention over the encoded condition tokens $C^g$. As positional encoding, we provide *absolute position tokens* $\mathcal{P}$, which provide for each token the absolute position in the sequence $S$. This sequence is used as additional input to the generative model. The output of the generative model in each step is one of the parametric probability distributions described in Eq. 8. Since the probability distributions for discrete and continuous values have a different numbers of parameters, we use a different final linear layer in the generative model for continuous and discrete parameters. Similar to the encoder, the architecture of the generative model is based on BART (Lewis et al., 2020).

**Training**: During training, we create a ground truth sequence $S^{\text{GT}}$ with randomly permuted objects. We generate the condition $C$ as a copy of $S^{\text{GT}}$ and mask out a random percentage of the tokens by replacing them with the mask token $\mathcal{M}$. The boundary encoder $g_\psi^\mathcal{I}$, the condition encoder $g_\phi$ and the generative model $f_\theta$ are then trained jointly, with the task to generate the full sequence $S^{\text{GT}}$. For unmasked tokens in $C$, this is a copy task from $C$ to the output sequence $S$. For masked tokens, this is a scene completion task. We use the negative log-likelihood loss between the predicted probabilities $p(s_i)$ and ground truth values $s_i^{\text{GT}}$ for tokens corresponding to continuous parameters, and the cross-entropy loss for the object category $\tau$. The model is trained with teacher-forcing.

**Sampling**: We generate a sequence auto-regressively, one token at a time, by sampling the probability distribution predicted by the generative model (as defined in Eq. 7) in each step. We use the same model for both conditional and unconditional generation. For unconditional generation, we start with a condition $C$ where all tokens are mask token $\mathcal{M}$. To provide more complete information about the partially generated layout to the encoded condition tokens $C^g$, we update the condition $C$ after each generation step by replacing mask tokens with the generated tokens. Empirically, we observed that this improves generation performance. An illustration and the full algorithm of this approach is shown in the supplementary. Once a layout has been generated, we populate the bounding boxes with objects from the dataset with a simple retrieval scheme. For each bounding box, we pick the object of the given category $\tau$ that best matches the size of the bounding box. In the supplementary, we present an ablation of the tokens $\mathcal{O}^i$, $\mathcal{R}^i$, and $\mathcal{P}$ that we add to the conditional encoder and generative model.

## 4 RESULTS

**Datasets**: We train and evaluate our model on the 3D-FRONT dataset (Fu et al., 2021). It consists of of about 10k indoor furniture layouts created by professional designers. We train on the BEDROOM category and follow ATISS preprocessing which removes a few problematic layouts that have intersections between objects, mislabeled objects, or layouts that have extremely large or small dimensions. For further details on the preprocessing, we refer the reader to ATISS (Paschalidou et al., 2021). This yields approximately 6k/224, 0.6k/125, 3k/516 and 2.6k/583 total/test set layouts for BEDROOM, LIBRARY, DINING, LIVING, respectively.

**Baseline**: ATISS (Paschalidou et al., 2021) is the most recent furniture layout generation method that provides the largest amount of control over the generated layout, and is therefore most related to our method. However, ATISS does not provide pretrained models, hence we train their models using the official code [1] matching their training settings as closely as possible. While ATISS does not support attribute-level conditioning, we can still use it as a baseline by applying the sampling procedure defined in Eq. 6: we sample tokens as usual, but when reaching a token that is given as condition (i.e. a token in $C$ that is not a mask token), we use the value given as condition instead of the sampled value.

---

[1]https://github.com/nv-tlabs/atiss, commit `0cce45b`

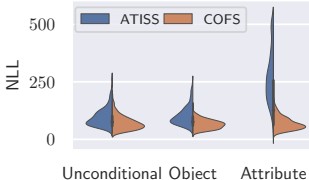
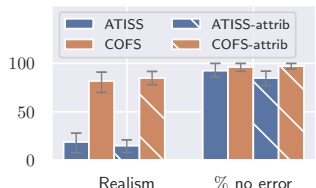

Figure 3: **NLL comparison.** We compare the NLL of our method to ATISS in three settings: (**Uncond.**) We measure unconditional generation performance as the NLL of BEDROOM test set layouts in our model. (**Object**) We measure the NLL of layouts generated with object-level conditioning and (**Attribute**) attribute-level conditioning. Note how our method performs slightly better than ATISS on unconditional generation and object-level conditioning, while showing a clear advantage in attribute-level conditioning.

Figure 4: **Perceptual Study**. We compare the percentage of comparisons in which users found either method more realistic, and the percentage of results in which users did not find any obvious errors such as object intersections. Results show a large advantage for COFS in realism of layouts generated with attribute-level conditioning (-attrib), and a smaller, but still significant advantage in the percentage of error-free layouts.. This advantage is also present in the unconditional setting.

### 4.1 QUANTITATIVE RESULTS

**Metrics**: For *unconditional generation*, we use the negative log-likelihood of test set sequences in our model as main quantitative metric. A small NLL shows that a model approximates the dataset distribution well. For both *object-level* and *attribute-level* conditioning, we use the NLL of the generated sequences as metric. This includes the condition tokens that come from the test set. If the generated layout does not harmonize with the condition, the NLL will be high. Additionally, we performed a perceptual study for attribute-level conditioning and unconditional generation (details are given below and in the supplementary material, respectively).

**Choosing conditions**: For *object-level* conditioning, we remove three random objects from each test set sequence to obtain condition sequences $C$. For *attribute-level conditioning*, conditions $C$ are obtained from test set sequences by replacing all tokens except size and position tokens with mask tokens, effectively conditioning on the sizes and positions of all objects, and letting the generator infer the types and orientations of all objects.

**Discussion**: Figure 3 shows NLL results on the BEDROOM category. For unconditional generation, we can see that we perform on par or slightly better than ATISS. We believe that our slight advantage here might be due a more fine-grained sequence representation of the layout on our side, which allows for more detailed attention. For object-level conditioning, our performance is slightly better than ATISS, again because of detailed attention. Our main contribution, however, lies in attribute-level conditioning, where we can see a clear advantage for our method. Since ATISS cannot look ahead in the sequence, any generated token cannot take into account future condition tokens. The bidirectional attention of our encoder enables look-ahead and gives the generator knowledge of all future condition tokens, giving us generated layouts that can better adapt to the supplied condition.

### 4.2 PERCEPTUAL STUDY

We conducted two perceptual studies to further evaluate the quality of generated furniture layouts compared to ATISS. One of the studies focused on unconditionally generated layouts and the other on layouts generated with attribute-level conditioning. For this purpose, we randomly sampled layouts from the BEDROOM layouts evaluated in the previous section for both COFS and ATISS. Subjects were shown a pair of layouts generated from the same floorplan boundary by COFS and ATISS, and asked three questions: which of the layouts looked more realistic, and for each of the two layouts, if it showed obvious errors like intersections. A total of 9 subjects participated in the unconditional study, and 8 subjects participated in the attribute-level study. More details about setup can be found in the supplementary. Figure 4 shows the results. We can see that the our method produces significantly more realistic layouts compared to ATISS. The error plots on the right-hand side show that this only in some part due to avoiding obvious errors such as intersections.

### 4.3 QUALITATIVE RESULTS

A few examples of furniture layouts generated with attribute-level conditioning are shown in Figure 1 and in Figure 5. See the captions for details. Figure 6 additionally shows how attribute-level conditioning can be used to perform sequential edits of a furniture layout.

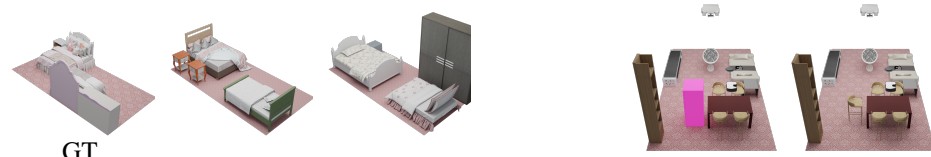

GT

Figure 5: **Attribute-level conditioning:** On the left, we show a GT floorplan. We set the condition to include two beds facing opposite directions and sample. The model generates two plausible layouts for this challenging case (see supplementary). On the right, we constrain the location of the next object to be sampled. The location is highlighted in pink. In this example, the network automatically infers the proper class and size. The constraints force the inferred size into a narrow range, such that the chair even matches the style of the chairs in the example on the left, even though we use a simple object-retrieval scheme.

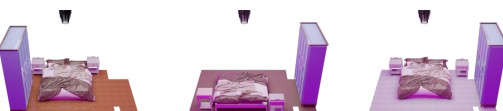

Figure 6: **Sequential edits with attribute-level conditioning:** We show how COFS can be used to selectively edit parts of a scene. Left shows GT and the other two are samples with classes and orientation as condition. When we change orientation of a few objects, COFS produces realistic layouts affecting only a part of the scene. More details in the supplementary.

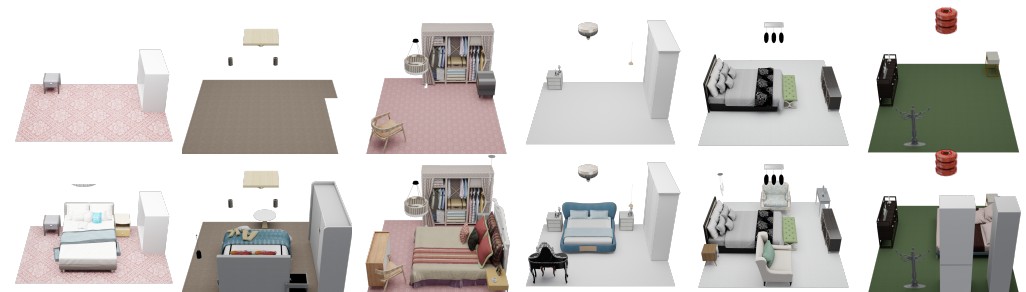

Figure 7: **Object-level conditioning.** In the top row, we show examples of object-level conditions that were used to condition generation of the scenes shown below. The generated layouts all plausibly combine the generated objects with the objects given as condition into realistic layouts.

In Figure 7, we show layouts generated with object-level conditioning, providing the objects shown in the top row as condition. Note how our method generates plausible layouts in each case.

### 4.4 OUTLIER DETECTION

We can also use COFS to perform outlier detection. To estimate the likelihood of each token, we follow (Salazar et al., 2020) and replace the token at $i$th position with [MASK]. This can be performed in parallel by creating a batch in which only one element is replaced with [MASK]. The likelihood of one object is then the product of likelihoods of all its attributes. Attributes or objects with low likelihood can then be resampled. Results on several of the layout categories of our dataset are shown in Figure 8. This can be thought of as a form of attributed-conditioned generation .

### 4.5 ADDITIONAL EXPERIMENTS WITH UNCONDITIONAL GENERATION

Here we present additional experiments with unconditional generation. We include two additional state-of-the art methods for unconditional generation, FastSynth (Ritchie et al., 2019) and Scene-Former (Wang et al., 2021), in our quantitative experiments.

We use a set of metrics that mostly derive from (Ritchie et al., 2019; Paschalidou et al., 2021). They are defined in greater detail in the supplementary. Following (Ritchie et al., 2019), we report the KL-divergence between the distribution of the classes of generated objects and the distribution of classes of the objects in the test set. We further report the Classification Accuracy Score (CAS) (Paschalidou et al., 2021). Additionally, we compute the FID by rendering the populated layout from a top-down view using an orthographic camera at a resolution of $256 \times 256$. We report the FID computed between

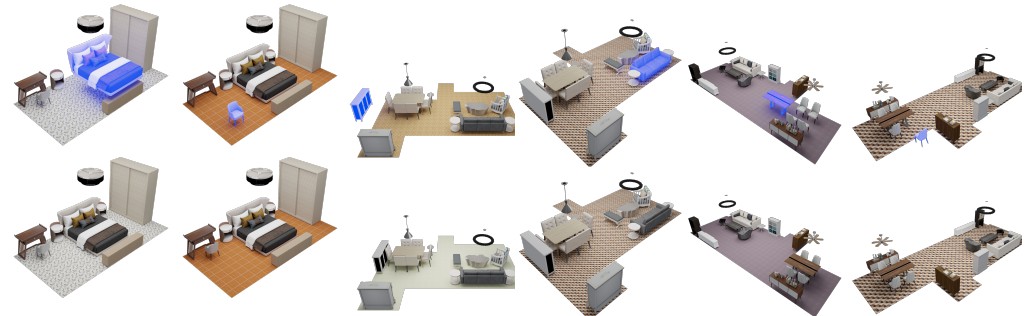

Figure 8: **Outlier detection:** Our model can utilize bidirectional attention to reason about unlikely arrangements of furniture. We can then sample new attributes that create a more likely layout. In contrast, ATISS can only sample whole objects. **Top row:** An object is perturbed to create an outlier (highlighted in blue). **Bottom row:** The object can be identified by its low likelihood, and new attributes sampled which place it more naturally.

these rendered top down images of sampled layouts and the renders of the ground truth layouts. Additional details are given in the supplementary.

Results are shown in Table 1. The results suggest that overall, COFS performs roughly on par or slightly superior to ATISS, with slightly inferior results in the CAS metric, comparable results in the FID metrics, and more substantially improved results in the KL-divergence metric. Examples of unconditionally generated layouts are shown in the supplementary.

Table 1: **Comparison on Unconditional Generation**: We provide floorplan boundaries from the Ground Truth as an input to the methods and compare the quality of generate layouts. We retrain the ATISS model and report metrics. The retrained model is called ATISS$^*$.

| | CAS $\times 10^2 (\downarrow)$ | | | | KL-Divergence $\times 10^3 (\downarrow)$ | | | | FID $(\downarrow)$ | | | |
|---|---|---|---|---|---|---|---|---|---|---|---|---|
| | BEDROOM | LIVING | DINING | LIBRARY | BEDROOM | LIVING | DINING | LIBRARY | BEDROOM | LIVING | DINING | LIBRARY |
| FastSynth | 88.3 | 94.5 | 93.5 | 81.5 | 6.4 | 17.6 | 51.8 | 43.1 | 88.1 | 66.6 | 58.9 | 86.6 |
| SceneFormer | 94.5 | 97.2 | 94.1 | 88.0 | 5.2 | 31.3 | 36.8 | 23.2 | 90.6 | 68.1 | 60.1 | 89.1 |
| ATISS$^*$ | 61.1 | **76.4** | **69.1** | **61.77** | 8.6 | 14.1 | 15.6 | 10.1 | **73.0** | 43.32 | 47.66 | **75.34** |
| Ours | **61.0** | 78.9 | 76.1 | 66.2 | **5.0** | **8.1** | **9.3** | **6.7** | 73.2 | **35.9** | **43.12** | 75.72 |

## 5   CONCLUSIONS

We proposed a new framework to produce layouts with auto-regressive transformers with arbitrary conditioning information. While previous work was only able to condition on a set of complete objects, we extend this functionality and also allow for conditioning on individual attributes of objects. Our framework thereby enables several new modeling applications that cannot be achieved by any published framework.

**Limitations and Future Work.**    We now discuss limitations of our model. The first is related to our simple object retrieval scheme based only on bounding box sizes. This often leads to stylistically different objects in close proximity even if the bounding box dimensions are only slightly different. We show such an example in the inset (left). 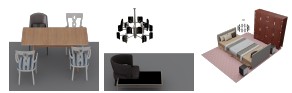
The second is related to the training objective of the model - we only consider the cross entropy/NLL. Thus, the network does not have explicit knowledge of design principles such as non-intersection, or object co-occurrence. This means that the model completely relies on the data being high-quality to ensure such output. In the supplementary, we highlight the fact that certain scenes in the dataset have problematic layouts, and our method cannot filter them out. We show an example of intersections in the inset (center). Thirdly, the the performance on the LIVING and DINING datasets is not as good as the other classes, which is clear from the CAS scores. This is in part because the datasets are small but also have significantly more objects than BEDROOM or LIBRARY. This leads to accumulated errors. We would like to explore novel sampling strategies to mitigate such errors. Lastly, while the conditioning works well, it is not guaranteed to generate a good layout. For example, in the inset (right), we set the condition to be two beds opposite each other, but the network is unable to place them in valid locations. Adding explicit design knowledge would help mitigate such arrangements, but we leave that extension to future work.

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
