# OpenReview forum: "COFS: COntrollable Furniture layout Synthesis"
_ICLR.cc/2023/Conference — Submitted to ICLR 2023_

### Official Review · Reviewer_CiDE · 2022-10-23

**Confidence:** 4
**Correctness:** 3
**Technical Novelty And Significance:** 2
**Empirical Novelty And Significance:** 3
**Recommendation:** 5

**Clarity, Quality, Novelty And Reproducibility:**

- the writing is clear, fluent and readable throughout

- the description of the proposed method is clear and explicit, with components being suitably motivated and described

- the proposed architecture is novel, and this is the first work that explicitly addresses the problem of layout synthesis with deep networks subject to attribute constraints (though note the preprint "Automatic Generation of Constrained Furniture Layouts" [Henderson 2017], which should perhaps be cited, and the fact that MCMC-based methods also support constraints)

- plenty of implementation details are given; however I encourage the authors to release the code to ensure full reproducibility


**Strength And Weaknesses:**

- authors identify a potential weakness of existing furniture layout generation methods, and propose a system that addresses it

- the proposed method is architecturally novel, utilising transformers like ATISS but in a significantly different configuration to support more sophisticated conditioning

- unconditional samples from the proposed model are no worse that those from ATISS

- attribute-conditioned samples have higher likelihood than those from ATISS, and the few qualitative examples given look reasonable

- a user study finds that both conditioned and unconditioned samples are significantly preferred (by a small group of humans) vs those of ATISS

- the qualitative results on fixing-of-outliers are a nice example of the power of this kind of approach

- there are insufficiently many qualitative results (in paper and supplementary) to convince me of the quality of the generations. Indeed, it is not clear whether the qualitative examples (even in supplementary) are curated or random. This could be mitigated by providing (e.g.) 100 uncurated samples per room type from each method

- the most critical results in the paper are those on attribute-level conditioning (since this is the main motivation, and the only feature that existing methods do not support). However, qualitative results are restricted to very few examples (fig. 5 + 6). This is insufficient evidence to support the key claims of the work. Moreover, some of these would be tractable with ATISS – either by rejection sampling (when the constraints do not restrict to an overly small subset of possible configurations), or (fig. 5, right) exhaustively finding the most-likely position, since there is only a single attribute and the likelihood is tractable. If rejection sampling ATISS in fact takes prohibitively long for fig. 5 (left) and fig. 6, then this should be stated and measured. I do however appreciate that the authors went to the effort of including a user study, including the attribute-conditional setting.

- it is not clear to me how NLL results with attribute conditioning are calculated for the baseline ATISS (which doesn't support such conditioning, hence the need for the proposed method). Please clarify this.

- there is no quantitative metric for outlier prediction / fixing; it seems like the former (at least) should be straightforward to measure quantitatively, roughly as a binary classification problem. This would avoid any impression that the figure just contains cherry-picked examples

- the negative log-likelihood (NLL) results in fig. 3 are given only for one room type. The full results should be given for all rooms (maybe in supplementary), to give a fair and balanced view of how well the two models perform.

- similarly, NLL is apparently only given when locations/sizes are conditioned on, not classes. This is unfortunate since I think a more likely practical application is (e.g.) "create a layout with three chairs", rather than "create a layout with arbitrary objects at these particular locations". Please consider adding these results. Also, quantitative evaluation in more diverse conditioning scenarios would be valuable – e.g. mixtures of position and class conditioning.

- fig. 7 should compare against ATISS, since that is noted to support the object-level conditioning task (i.e. completing partial layouts). In this case, a handful of uncurated samples from each method should be given

- the introduction clearly states that attribute-conditional generation is impossible (or at best very computationally expensive) with existing methods like ATISS. However, it doesn't give a convincing motivation of why attribute-conditioned generation is actually useful – i.e. in what practical use-cases one might have a complex constraint on classes and/or positions, but one wouldn't simply build the scene by hand

- some of the cited 'pre deep learning' methods do in fact support the kind of conditioning described in the paper. In particular, it is straightforward within an MCMC sampling scheme to fix certain variables (position of one object, etc.), resulting in a posterior sample that correctly accounts for this constraint. This should be made clear, as currently the paper reads like it is the first to support such conditioning.

- fig. 5 caption advertises the fact that the sampled objects are of a style that matches the ground-truth. However, from the method description this must simply be a coincidence, since the method only predicts bounding boxes, not particular object shapes / instances! Also, there aren't actually any chairs in the left scene, so far as I can see – maybe this comment refers to the right-hand one?

- as noted in the limitations section, and in common with related works, the method is limited in terms of output quality by the quality of the ground-truth dataset (3D-FRONT), which is well known for containing object intersections and other issues. While it may seem unfair to raise this as a criticism of the present work, it means that generated scenes are not suitable for most practical uses – a fact that has to weigh slightly negatively given the rather 'applied' focus of the paper.

- there does not appear to be any analysis of whether the model is memorising training layouts. This should be measured, given that relatively small size of 3D-FRONT compared with the relatively large transformer model (I appreciate it's smaller than ATISS, which is a good thing, but still there are millions of parameters, and overfitting is a potential issue). This could be resolved fairly easily by doing a search for nearest neighbors (in the training set) to a large set of sampled layouts.

- there is no test of statistical significance for the user study results. This is particularly problematic given the small number of participants. Relatedly, for the "326 responses" in the user study, is this 326 individual comparisons, or 326 x 24 comparisons, or something else?


**Summary Of The Paper:**

Authors propose a model for generating layouts of furniture in rooms. Unlike existing works in this area (e.g. ATISS), the proposed method allows conditioning on combinations of attributes (e.g. "place a chair somewhere, and some other object at this particular location"). The proposed model is a novel combination of transformer encoder and decoder, allowing arbitrary attributes of arbitrary objects to be fixed, and retaining permutation invariance. Evaluation is conducted on the 3D-FRONT dataset, including various metrics and a perceptual user study.

**Summary Of The Review:**

The paper addresses a clear gap in functionality of layout generation models, and proposes a sensible and novel approach. However, the empirical evaluation is currently marginally insufficient to convince me that the method indeed solves this problem well.

---

> ### Author Response · Authors · 2022-11-18
> **Rebuttal**
>
> Thank you for the detailed comments and suggestions. Below we address the main issues:
>
> # Qualitative Results and Comparison to ATISS for Attribute-Level Conditioning.
> We added several qualitative results to the supplementary material showing additional applications of our approach that are infeasible with existing methods. Specifically, we show qualitative results for fully automatic generation of furniture layouts that satisfy a set of detailed constraints (Figure 2, supplementary) and furniture placement suggestions that assist artists in semi-automatic layout creation (Figure 1, supplementary). Additionally, in Figure 8, we added a performance comparison to ATISS on attribute-level conditioning, showing that rejection sampling with ATISS quickly becomes infeasible as the number of conditions increases.
>
> # NLL for ATISS attribute-level conditioning with ATISS?
> The sampling procedure we use for ATISS is described in Section 4, paragraph ‘Baseline’:  when generating a token with ATISS that is part of the condition, we replace the generated token with the corresponding condition token. The NLL is then obtained by evaluating conditionally generated sequences in the learned model (ATISS or COFS), including the tokens that were copied from the condition. This measures how well the condition fits with the generated tokens. If the condition is a bad match for the generated tokens, the NLL for the sequence will be high. We will clarify this.
>
> # Quantitative Evaluation of Outlier Detection
> We can add a quantitative evaluation with synthetic outliers to the final version if requested. However we would like to point out that outlier detection in general works very well - in fact better than all other applications.
>
>
> # NLL for other room types
> Similar to previous methods, we chose bedrooms for the quantitative evaluation, as this category has the largest amount of data. Other categories are less reliable for a quantitative evaluation, as we run into mitigating issues that are present in the limited-data regime.
> # NLL for other types of conditions
> We agree that additional evaluations with different types of attribute-level conditions could be interesting to see, however, we believe that these are not strictly necessary to demonstrate the advantage of our attribute-level conditioning over ATISS, which is already shown clearly in Figure 3 (supplementary) . We have still added a new form of conditioning in the supplementary (Figure 2). We can however add more experiments if requested.
>
> # Practical Use-Cases of Attribute-Level Conditioning
> Attribute-level conditioning enables several applications towards fully automatic creation of interior spaces in large virtual worlds, like fully automatic generation of furniture layouts that satisfy a set of detailed constraints (for example, generating N different furniture layouts where a desk faces the bed), or furniture placement suggestions that assist artists/designers in semi-automatic layout creation. We have added a discussion in the supplementary. We added several examples of these applications in Figures 1 and 2 of the supplementary and a comparison to the performance of ATISS on these applications in Figure 8, of the supplementary.
>
> # Some pre-deep learning methods support attribute-level conditioning
> We will clarify that we are the first deep learning method to support this for furniture layouts.
>
> # Figure 5 Mentions Matching the Style of Chairs
> We wanted to highlight that in the right-hand example, the class and size of the object is predicted accurately enough for the retrieval to return the same model as the other chairs. We added text to better convey our intentions.
>
> # Evaluation of Overfitting is Missing
> We added Fig 2 (in the supplementary) where we show the nearest training-set neighbors of a few generated scenes, showing that the generated scenes are novel, especially when changing the conditions.
>
> # Confidence Intervals in User Study
> We do have some tests of significance, and the bars on the user study actually are the 95% Confidence Intervals
> We will clarify that the error bars in Figure 4 (main paper) show 95% confidence intervals.

---

> > ### Author Response · Authors · 2022-12-10
> > **Additional results**
> >
> > We added some more results as an extension of Fig 1 (supplementary) to
> >
> > https://sites.google.com/view/cofssupplementary
> >
> > where we condition only on bed *class* and *location* in by unmasking those attributes (Bed location shown in white.).  This type of conditioning on partial attributes is not possible in ATISS, unless for some specific combination of attributes.

---

### Official Review · Reviewer_iswK · 2022-10-25

**Confidence:** 3
**Correctness:** 4
**Technical Novelty And Significance:** 2
**Empirical Novelty And Significance:** 3
**Recommendation:** 6

**Clarity, Quality, Novelty And Reproducibility:**

clarity : it is clear.

quality : good quality.

novelty : not so much, fairly "standard".

reproducibility : good, the supplementary is comprehensive,

After reading the response I'm keeping the score.

**Strength And Weaknesses:**

strength: this paper is well presented, the method is clear to read, and the evaluation is solid. Using data likelihood and omitting either objects / attributes and having the model regenerating them is a good way to evaluate, and it performs better than the baseline. I especially liked the user study on realistic-ness, where we can see COFS generates more realistic scenes rated by end-users -- this is gold standard.

weakness: the technical aspect of this work (by nature) isn't very novel. A naive interpretation is simply this model can be conditioned by a richer context (object + attribute) as opposed to objects alone. Therefore, one would like to see more works on down-stream usage of the system. The proposed system can be used as an augmented design tool, where designers can interact with the system to create, or re-create realistic scenes. This is more of a HCI / Siggraph aligned work, and I wonder if ICLR isn't quite the right fit for it.

An amazing user study would be give designers a picture, and ask them to re-construct it in 3D using your tool. How might this process look? I feel data on interacting with these generative tools are vastly lacking, but is extremely valuable as that is their ultimate use-case.

**Summary Of The Paper:**

This paper improves upon prior approaches that generates furnitures. The proposed method is more controllable, able to generate / complete a scene from either object-level conditioning (think rendered, physically placed objects) or attribute-level conditioning (think word, stylized language description of the scene). The evaluation is thorough.

**Summary Of The Review:**

I personally like this paper. It addresses a realistic problem, and it is well evaluated and solid. My only concern is this work might not be the right fit for ICLR.

---

> ### Author Response · Authors · 2022-11-18
> **Rebuttal and kickstarting discussion**
>
> Thank you for the detailed comments and suggestions. Below we address the main issues:
>
> # Novelty
> COFS is the first furniture layout generator to support efficient attribute-level conditioning. This enables several applications towards fully automatic creation of interior spaces in large virtual worlds, like fully automatic generation of furniture layouts that satisfy a set of detailed constraints (for example, generating N different furniture layouts where a desk faces the bed), or furniture placement suggestions that assist artists in semi-automatic layout creation. Technically, this is enabled by our generation approach with masked sequences and a transformer encoder that provides detailed information about the condition. To our knowledge, both generation with masked sequences and a transformer encoder have not been applied to furniture layout generation before. We added several examples of the applications we mentioned to Figures 1 and 2 of the supplementary material and a comparison to the performance of ATISS on these applications in Figure 8.
>
> # More Appropriate for HCI venues than ICLR?
> While our method could have some downstream applications that may be interesting to the HCI community, like an AI-assisted design tool, the core of our method concerns conditional generation of 2D/3D layouts, being the first to use masked language models for improving conditioning performance of set-like data (layouts) which seems like an appropriate topic for ICLR. Additionally, we would like to point out that one interesting class of downstream applications besides AI-assisted tools for designing individual layouts, could be automatic generators for large virtual worlds (i.e. creating large numbers of 2D layouts without direct user interaction) that can be controlled through constraints that are given in advance. We give some examples of applications in the supplementary Section B1.

---

### Official Review · Reviewer_6pUr · 2022-10-25

**Confidence:** 3
**Clarity, Quality, Novelty And Reproducibility:** The originality of the work is arguab…
**Correctness:** 3
**Technical Novelty And Significance:** 2
**Empirical Novelty And Significance:** 3
**Recommendation:** 5

**Strength And Weaknesses:**

Strenghs:
- The problem being studied is important. The authors point out the weakness of existing method that they are usually not flexible enough.
- The generated scenes look realistic.

Weaknesses:
- The presentation of the paper could be improved. For example, in Sec 3.2, it is unclear whether each s_i corresponds to B_i in Sec 3.1. In Sec 3.3, it is unclear what is the exact form of C.
- Overall, the method is close to ATISS, hence the novelty could be limited. As mentioned by the authors, ATISS can also generate scenes conditioning on object attributes by the replacement technique (Sec 4, baseline). The overall mechanism is similar, and the proposed method is also auto-regressive.
- The performance of the method is close to ATISS in unconditional generations. For conditional generations, it is difficult to tell if the advantage of the proposed method still holds if ATISS also does attribute conditioning during training.

**Summary Of The Paper:**

This paper proposes an indoor scene generation algorithm that supports conditioning on a set of object attributes. By randomly permuting furniture objects at training time, the method is trained to be approximately invariant to object permutations. A transformer encoder is added to provide cross-attention over the complete conditioning information in each step. The experiments show that the proposed method generates realistic scenes in both conditional and unconditional cases.

**Summary Of The Review:**

Overall I feel the paper is not fully ready considering its presentation and novelty. I would like to hear the authors' response on these comments.

---

> ### Author Response · Authors · 2022-11-18
> **Rebuttal**
>
> Thank you for the detailed comments and suggestions. Below we address the main issues:
>
> # Unclear presentation
> Note that s_i is one element of the sequence S, thus it includes the SOS and EOS tokens as well and does not directly correspond to B_i. We will mention this more explicitly.
> The condition sequence C is defined in Section 3.2 as a partial sequence, i.e.a subset of S, with some tokens replaced by mask tokens. The ‘training’ paragraph in Section 3.3 describes how it is constructed at training time from ground truth sequences.
>
> # Novelty
> COFS is the first furniture layout generator to support efficient attribute-level conditioning. This enables several applications towards fully automatic creation of interior spaces in large virtual worlds, like fully automatic generation of furniture layouts that satisfy a set of detailed constraints (for example, generating N different furniture layouts where a desk faces the bed), or furniture placement suggestions that assist artists in semi-automatic layout creation. Technically, this is enabled by our generation approach with masked sequences and a transformer encoder that provides detailed information about the condition. To our knowledge, both generation with masked sequences and a transformer encoder have not been applied to furniture layout generation before. We added several examples of the applications we mentioned to Figures 2 and 3 of the supplementary material and a comparison to the performance of ATISS on these applications in Figure 8.
>
> # Performance Compared to ATISS
> We added two additional evaluations that compare to ATISS in the supplementary material: Figure 8 shows a performance comparison of our approach to ATISS, where ATISS uses rejection sampling to support attribute-level conditioning. Note that for rejection sampling, the complexity increases exponentially with the number of attribute-level conditions. Results show a clear advantage for our approach. Second, in Figure 3, we visualize the probability distribution for the location of other class when conditioning on a given bed location and compare our distribution to ATISS. ATISS shows flat and uncertain distributions. This is due to the inability of ATISS to look ahead and take the orientation attribute of the desk into account. In COFS, the predicted probability distribution for the location is conditioned on the given conditions and only shows maxima in locations that admit the given constraints.

---

### Official Review · Reviewer_J17x · 2022-10-25

**Confidence:** 5
**Correctness:** 4
**Technical Novelty And Significance:** 2
**Empirical Novelty And Significance:** 3
**Recommendation:** 6

**Clarity, Quality, Novelty And Reproducibility:**

### Clarity
The paper is clear and easy to follow

### Quality
The work is well executed and the evaluations are quite adequate.

### Originality
Amount of novelty is quite limited: largely follow the pipeline of the prior works. The new conditioning mechanism has also been attempted, as far as I know, in other tasks.

### Reproducibility
Should be - largely relies on the codebase of prior works and the new ideas seem to be easily implementable

**Strength And Weaknesses:**

# Strengths
- Simple yet effective idea that extends existing autoregressive scene synthesis models and enable more flexible applications.
- Good empirical performance: although the performance w.r.t the three distributional similarity metrics is not too much different from ATISS, the perceptual study results are much better. I definitely think the human perceptual study tells much more in this case.

# Weaknesses
### Novelty
- Amount of novelty is rather limited, both on the scene synth side and on the set generation side.

### Motivation & Design Choices
- Some additional discussions regarding the positional encoding tokens would be nice. I get that they helps training, as shown in the supplementary, but I am unsure what's the intuition behind these, especially the absolute position tokens on the decoder side.
- I think it might make sense to highlight the difference between "what was not possible with prior work" versus "what was possible, but only in a very complicated way". I get that the conditioning in this work can be very flexible, but it appears to me that most of the stuff can also be accomplished by prior works, by following a "traditional" sequence and hardcode some of the info when needed. Would be more helpful if the authors can highlight something unique that only this work can do, and showcase these a bit more.

### Evaluation
- My main issue lies with the set of evaluations the authors chose to perform. In the case of this work, where main novelty lies in the ability to enable more flexible forms of conditions, I do not think overall scene quality is the right metric to use here. Granted, these metrics are important as they show that the method generates reasonable outputs, they do no highlight the benefits of the new ideas in this paper as much. There's a few qualitative results in the supplementary that showcases such flexibility, however, I think it more evaluations on this respect are needed. A few ideas here: 1. visualize the distributions for a single token between ATISS and COFS, and show that the extra conditioning of COFS actual makes these distributions more reasonable; 2. A user study that task someone to edit a scene given a natural language instruction, and hopefully demonstrates that COFS allows a better editing experience; 3. A more fine-grained quantitative analysis showing that performance of ATISS decreases further when the form of conditioning is more complex/irregular.
- I am slightly concerned with the perceptual study setup - judging from Fig 2 in the supplementary, it seems that it might be hard for the participants to make good decisions due to the choice of camera angles and the rendering quality.

### Misc
- Figure 5: "even matches the styles of the chairs": IIUC the models are retrieved based on bbox dimensions, so a claim regarding styles is probably not the best one here? (it's probably a byproduct of predicting a bbox of exactly the right size?)

**Summary Of The Paper:**

This paper extends existing transformer-based autoregressive indoor scene synthesis method by introducing a new conditioning mechanism that allow the input sequence to contain, at any location, an arbitrary number of masked tokens, whose values are to be predicted. Three set of positional encodings are introduced to enable this form of conditioning: object index and attribute types for the encoder, and global absolute positions for the decoder. It is shown that the proposed method performances better than prior works, especially with respect to human perceptual studies. Limited discussions are provided that demonstrate that the proposed framework is more flexible thanks to the conditioning mechanism.

**Summary Of The Review:**

I have mixed feeling about this work. On one side, I am impressed by the quality of the results and really like the amount of flexibility provided by this new conditioning formulation. On the other side, the amount of novelty in this work is rather limited - both the transformer based scene synth part and the set generation / arbitrary conditioning part. My rating of this paper would have been higher if the authors provide more fine-grained analysis on what *exactly* does the new ideas bring. However, since the current evaluation focuses on overall quality, which does not necessarily origin from the novel (in my mind) part of the paper, I am hesitant to recommend acceptance and lean towards rejection for now.

=====Post Rebuttal Comment=====

Thanks for the response to my concerns. I appreciate the clarification regarding the experiment setup and the positional tokens. The additional experiments do showcase the strength of this method, though I still don't think it showcases scenarios where previous method *cannot* work.

I thus still am quite ambivalent about this paper. However, after the reviewer meeting, I do agree with the AC and other reviewers that audiences of this conference can get some inspiration should this work be accepted. I was a bit hesitant in arriving at this conclusion because I might be quite biased as I have worked on related problems for a long time. However, the discussion did sway me slightly towards the acceptance side and I thus raise my score from a 5 to a 6. I still won't champion for this paper, and am fine with it going either way.

---

> ### Author Response · Authors · 2022-12-10
> **Post-rebuttal score**
>
> Thank you for your message. Glad to know that our additional experiments helped in clarifying some of the loose ends. Happy to hear that there is support from the reviewers and the AC! Indeed, we agree that ATISS can also be used to produce ‘conditional’ results, just that it is unwieldy as shown in Figure 8. Given a chance, we will do our best to further clarify the advantages in writing. Thank you again for making the post-rebuttal discussion open and informative.

---

### Decision · Program_Chairs · 2023-01-20

**Decision:**

Reject

**Justification For Why Not Higher Score:**

Overall reviewers were not that excited about this work.  They felt the novelty was limited and that it was perhaps more appropriate for a graphics or vision conference rather than ICLR.  The AC agrees that the novelty to the ICLR audience is limited and the work would be better targeted to the graphics and vision communities.


**Justification For Why Not Lower Score:**

Like the reviewers, the AC also likes scene generation and believes that perhaps acceptance will help encourage more people in the ICLR community to think about useful representations and approaches for scene-generation.   The approach and paper was also solid and a clear, if not exciting, improvement over prior work.


**Metareview: Summary, Strengths And Weaknesses:**

Summary: The paper proposes a method for object or attribute-conditioned indoor scene synthesis (furniture placement) with a transformer-based encoder-decoder.  This allows for more flexible constraints than prior work.  The key idea is to specify the conditioning as a input sequence (which encodes objects and their attributes) that can include masked tokens (for unconstrained attributes).  The model is based on BART (a masked language model) and is trained to be permutation invariant by having permuted objects during training.  Experiments on the 3D-FRONT dataset, including a perceptual study, show that the proposed approach generate better layouts than prior work (ATISS).

Strengths:
- The proposed method is the first to use mask-language-model for scene generation and the first deep learning based scene synthesis approach to handle conditioning on attributes
- Proposed method appears to work well
- The paper is well-written with a clear and explicit description of the method

Weaknesses:
- There are weaknesses in the evaluation (note that this is a common weakness with prior work in indoor scene generation as well)
- The technical contribution is limited as it is similar to ATISS but with masking
- The work may be of limited interest to the broader ICLR community and may be a better fit for graphics or vision conference


**Summary Of Ac-Reviewer Meeting:**

The AC met with two reviewers (J17x, CiDE) who both expressed they like scene generation, thought the paper was borderline, and were okay with either accept or reject.  The reviewers were not sure if the ICLR community would benefit from this work as it has limited technical novelty.  The AC expressed the opinion that the ICLR audience can potentially benefit from the work as an interesting problem that could inspire the learning community to devise improved representation and methods.

The other two reviewers were not able to make the meeting and discussed on OpenReview. Reviewer iswK indicated that they were okay with rejecting or accepting (their rating was a 6), and the fourth reviewer (6pUr) was comfortable with rejecting.